# Involuntary ambulatory triage during the COVID-19 pandemic – A neurosurgical perspective

**Harald Krenzlin**[1]*, **Christoph Bettag**[2], **Veit Rohde**[2], **Florian Ringel**[1], **Naureen Keric**[1]

**1** Department of Neurosurgery, University Medical Center of the Johannes Gutenberg-University, Mainz, Rhineland Palatinate, Germany, **2** Department of Neurosurgery, University Medical Hospital of the Georg August University, Göttingen, Lower Saxony, Germany

* harald.krenzlin@unimedizin-mainz.de

## Abstract

### Background

The coronavirus disease 2019 (COVID-19) pandemic poses an unprecedented challenge to health-care systems around the world. As approximately one-third of the world´s population is living under "lockdown" conditions, medical resources are being reallocated and hospital admissions are limited to emergencies. We examined the decision-making impact of these actions and their effects on access to hospital treatment in patients with neurosurgical conditions.

### Methods

This retrospective cohort study analyzes hospital admissions of two major neurosurgical services in Germany during the nationwide lockdown period (March 16th to April 16th, 2020). Spinal or cranial conditions requiring immediate hospital admission and treatment constituted emergencies.

### Results

A total of 243 in-patients were treated between March 16th and April 16th 2020 (122 patients at the University Medical Center Mainz, 121 patients at the University Medical Center Göttingen). Of these, 38.0±16% qualified as emergency admission. Another 1,688 admissions were reviewed during the same periods in 2018 and 2019, providing a frame of reference. Overall, emergency admissions declined by 44.7±0.7% during lockdown. Admissions for cranial emergencies fell by 48.1±4.44%, spinal emergencies by 30.9±14.6%.

### Conclusion

Above findings indicate that in addition to postponing elective procedures, emergency admissions were dramatically curtailed during the COVID-19 lockdown. As this surely is unexpected and unintended, reasons are undoubtedly complex. As consequences in

**Data Availability Statement:** All relevant data are within the manuscript and its Supporting Information files.

**Funding:** The author(s) received no specific funding for this work.

**Competing interests:** The authors have declared that no competing interests exist.

morbidity and mortality are still unpredictable, efforts should be made to accommodate all patients in need of hospital access going forward.

## Introduction

At the beginning of 2020, a novel Coronavirus (2019-nCoV) contagion had spread to 18 countries, four of them reporting human-to-human transmission. The World Health Organization (WHO) subsequently declared the outbreak a Public Health Emergency of International Concern (PHEIC) on January 30. The first cases of respiratory infections due to 2019-nCoV were reported on December 31, 2019, occurring in China's Hubei province. [1]

Similarity to the severe airway respiratory syndrome (SARS) prompted a later change in nomenclature from 2019-nCoV to SARS-CoV-2. [2] Meanwhile, the growing coronavirus disease 2019 (COVID-19) pandemic had presented a new and unprecedented challenge to our healthcare system. [3] Hospitals and intensive care units are near breaking points in dealing with those suffering from this infection, forcing lockdown conditions on one-third of the world´s population. [4] As of April 19th, 2020, nearly 2 million people had been infected worldwide. In Germany the number of confirmed cases had risen to 145,742 with 4,642 deceased.

Aside from guidelines issued by the WHO, each country was separately tasked with handling the SARS-CoV-2 viral threat. In Germany, the chief governmental scientific institute in the field of biomedicine and the most important body for safeguarding public health is the Robert Koch-Institute (RKI). Its pandemic plan, which was revised and then amended in March 2020, offers specific protocols for addressing situations like the current SARS-CoV-2 outbreak. The strategy to deal with evolving epidemiological phases is marked by three fluid levels of escalation: containment, protection, and mitigation. We are currently at the point of pandemic mitigation, having enacted a nationwide lockdown on March 16, 2020. This included social distancing, curfews, and a ban on public assembly, with the closure of shops, schools and childcare. A televised address by Chancellor Angela Merkel was delivered 2 days thereafter to reinforce these measures.

To prevent a healthcare system overload, such as that occurring in Italy, hospitals hurriedly adapted to the state of pandemic. The German Hospital Federation, in collaboration with other hospital-related federal organizations and representatives of larger medical centers, agreed to increase ventilator and intensive care capacities. At the same time, nursing staff medical and medical assistants were duly instructed and reallocated to offset expected increases in patients stricken with COVID-19. A unified decision to halt elective hospital admissions and postpone non-emergency surgeries made this historic restructuring possible. The task of triage was assigned to medical personal at all tiers of the German healthcare system. As neurosurgical procedures have declined, however, the ramifications of these policies on pertinent decision-making and inpatient treatment access remain unclear.

This study was conducted to analyze likely impediments to receipt of adequate neurosurgical treatment during the COVID-19 pandemic.

## Methods

### Patient population

All admissions to the Department of Neurosurgery, University Medical Center Mainz between January 1st to April 19th, 2018–2020 were reviewed for this study. Admission type (elective vs.

emergency) and means (transfer from outside hospital, in- hospital transfer, emergency room visit, or referral) were recorded.

The time span beginning March 16[th] to April 19[th], 2020 was defined as core lockdown period. Emergency admissions and means (transfer from outside hospital, in- hospital transfer, emergency room visit, or referral) during this time were assessed and compared with the same timeframe in 2018 and 2019. Two major neurosurgical departments in Germany; The Department of Neurosurgery at the University Medical Center of the Johannes Gutenberg University Mainz and The Department of Neurosurgery at the University Hospital of the Georg August University Göttingen contributed to this analysis.

## Emergencies

Emergencies were defined as those requiring immediate or timely access and treatment within 1–2 days. Each was then categorized by admitting diagnosis as either cranial or spinal emergency. Cranial emergencies were further stratified as vascular, tumor-related, traumatic or infectious and miscellaneous; designating spinal emergencies as traumatic, tumor-related, degenerative or infectious.

## Statistical analysis

All clinical parameters were collected and analyzed retrospectively. Data acquisition and analysis proceeded in anonymous fashion and is expressed as mean ± standard deviation (SD). Student's T-test (two tailed) and two-way analysis of variance (ANOVA) with Tukey's multiple comparison post hoc test were performed using GraphPad Prism version 8.4.2 for macOS, GraphPad Software, La Jolla California USA, www.graphpad.com. A value of $P < 0.05$ was accepted as statistically significant.

## Ethical approval

Data acquisition and analysis was performed in an anonymous fashion and was approved by the Ethics Committees of the medical association of Rhineland Palatinate and Lower Saxony, Germany. According to the local laws, no informed consent is necessary for such kind of analysis.

## Results

From January 1[st] to April 19[th], 2020, a total of 822 patients were admitted to the Department of Neurosurgery, University Medical Center Mainz. Mean patient age was 57.7±16.9 (range 16–96 years), similar numbers were recorded in 2018 and 2019 (2018: 856 patients, 57,4±17.6 years; 2019: 968 patients, 59,6±17.0 years). A total of 101 emergency admissions (spinal: 28; cranial: 73) took place as emergency during March 2020. In the 2 weeks prior to lockdown, there were 66 emergency admissions (spinal: 17; cranial: 49). During the first 2 weeks of lockdown, emergency admissions sharply declined to 30 (cranial: 24, spinal 11) and remained low, totaling 38 (cranial: 28, spinal: 10) throughout the beginning of April. Overall emergency admissions dropped by 47% (cranial: 51%, spinal: 35.3%) during the first 2 weeks of lockdown and by 62.1% (cranial: 63.3%, spinal 58.8%) during weeks 3 to 4. (Fig 1). Vascular emergencies showed the steepest decline (81.8%), followed by admission for trauma (53.3%) (Fig 1). Patient routes leading to hospital admission likewise changed during the COVID-19 pandemic. Referrals, hospital transfers and visits to the emergency department all declined during the first 2 weeks of shut down and even further through week 3 to 4. (Fig 1)

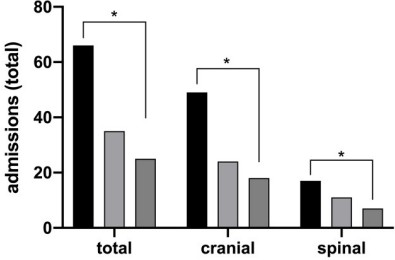
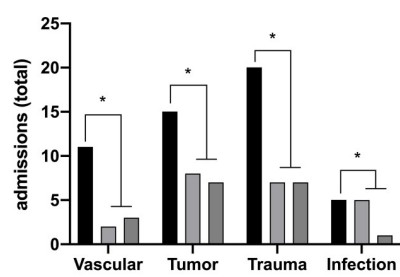
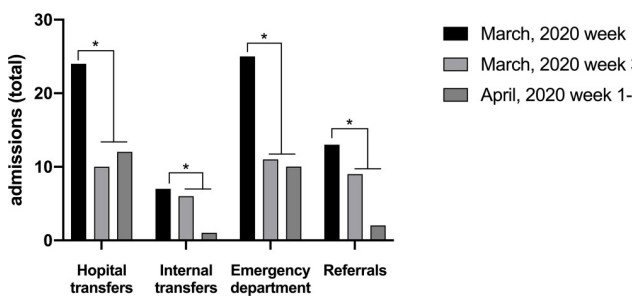

**Fig 1. Emergency admissions to the Department of Neurosurgery at the University Medical Center Mainz 2 weeks before and during the nationwide lockdown.** A) Admissions for spinal and cranial emergencies. B) Distribution of cranial emergencies. C) Patient routes leading to hospital admission during the COVID-19 pandemic.

A 5-weeks period between March 16[th] to April 19[th], 2020 constituted the core lockdown phase. Emergency admissions at both departments declined by 44.7±0.7% (cranial: 48.1 ±4.44%, spinal 30.9±14.6%) compared to 2018 and 2019 (p < .05) (Fig 2).

Combined, admissions due to brain tumors declined by 61.1±38.8%, those for traumatic brain injury by 53.8±0.6.% and those for 42.0±39.%. Both hospitals saw the highest decline of admissions via the emergency department 53.1±5.7%. Emergency transfers from smaller hospitals remained unchanged, going down by 3.2±7.9% compared to previous years.

During the lockdown period, the ICU bed capacity, availability of personal protective equipment (PPE) and manpower for the treatment of COVID-19 patients has never been fully exploited through treating those admitted with SARS-CoV2 infection in both Medical Centers.

## Discussion

The present analysis indicates that in addition to postponing elective procedures during the COVID-19 crisis, emergency admission dramatically declined. To our knowledge, this is the first documentation of the impact made by consecutive epidemiological countermeasures enacted for non COVID-19 emergency admissions to hospitals in Germany during the SARS-CoV-2 pandemic.

The reservations carried by the general public against hospital visits during the pandemic are reflected by plummeting numbers of visits to the emergency department. Our observations are aligned with reported analogous declines of patient admissions in Italy for minor strokes and transient ischemic attacks during the pandemic spread. [4] Moreover, patient arrivals were increasingly too late to initiate acute stroke treatment, causing reporting rates of thrombolysis and combined thrombolysis-thrombectomy to decline. [5] So far, it is unknown what

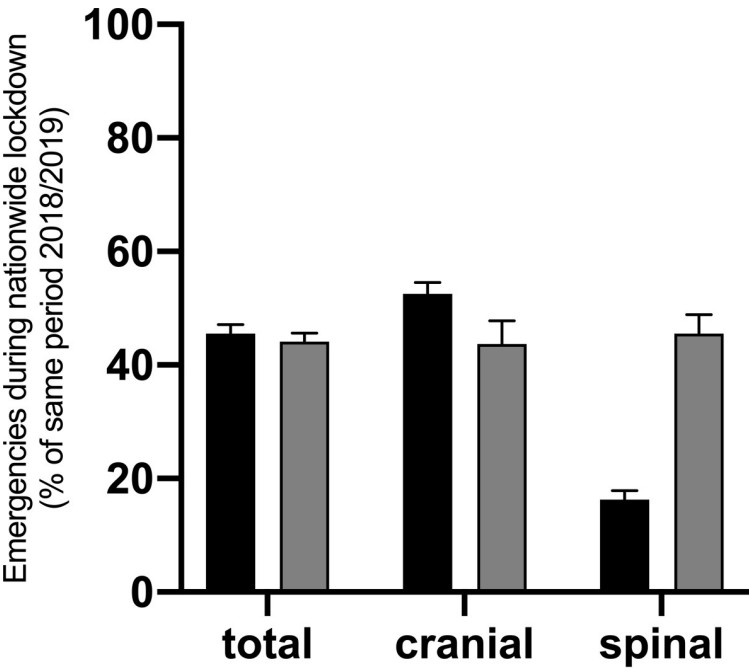

**Fig 2. Combined emergency admissions to the Departments of Neurosurgery of the University Medical Centers Mainz and Göttingen during lockdown compared to 2018/2019.**

kept people away from hospitals and if this tendency was more pronounced in larger medical centers. One reason might be the overall attempt to avoid exposition to patients with SARS-CoV2 infection perceived as unnecessary by avoiding hospitals and doctor consultations in situation, where such would have been commonplace previous to the pandemic. In turn, this refusal might lead to protracted hospital admissions or an increase of prehospital mortality. Countries participating in EuroMOMO (European mortality monitoring activity; https://www.euromomo.eu/) reported high above average excess death rates coinciding with the SARS-CoV2 pandemic. As the excess mortality only partially reflects the COVID-19 case fatality, delayed hospital admission might be a contributing factor. Measures taken to mitigate the risk of viral spreading such as social distancing and refrain from visiting the elderly lead to delays in detection of situations where those who we aim to protect from infection are unable to obtain the help they require due to conditions other than COVID-19. As in Italy, the prevailing inclination of Germans overall and among primary care physicians in particular was to avoid admitting elderly patients or those with multiple comorbidities or cancer. [6] The tenets of COVID-19 risk profiling are unfortunately often shared by patients admitted for neurosurgical vascular or spinal emergencies. In the absence of reasonable explanations why the

incidence of malignancies or vascular incidents should decline during a viral pandemic, it seems that patients in need of neurosurgical care are either admitted delayed or not at all during this difficult time. It remains to be seen whether the decline of admissions for traumatic brain injury is rather caused by a lower incidence due to restricted mobility or an epiphenomenon of social distancing leading to isolation and unobserved accidents with a higher pre-hospital cases fatality. Additionally, delays in diagnosis and treatment of non-life-threatening conditions might lead to an increased morbidity burden in the month to come.

As a result, transparency in communication and avoidance of unnecessary ambiguity on the state of our hospitals and the feasibility of emergency treatment is utmost important. It is therefore of exceptional importance to minimize errors in obligatory patient prioritization during the COVID-19 pandemic. [7] In a joint effort the German Society of Neurosurgery (DGNC) and the Professional Association of German Neurosurgeons (BDNC) thus jointly drafted a paper delimiting all non-elective surgical interventions in their purview (https://www.dgnc.de/fileadmin/media/dgnc_homepage/publikationen/downloads/DGNC_BDNC_non-elective_surgical_interventions.pdf) to rectify the situation. Ultimately, any underlying condition (cancer, organ failure, vascular disease, secondary complications of a disease or accident) that may inflict irreversible harm if prolonged qualifies as a time-sensitive vulnerability and should still be considered for immediate treatment, despite the ongoing pandemic. [8] Additionally, patients with cancer are at higher risk of COVID-19 infection and have poorer outcomes than the general population. [9] In cases of emergency, the results of COVID-19 testing may not be available prior to surgery so that particular precautions to minimize possible exposure to the novel coronavirus during the procedures are mandatory. [10] During times of unprecedented strain on our healthcare system, an attempt to balance the acuity of medical needs with available resources is of overarching importance. [8, 11] It is noteworthy that none of the cases primarily postponed as elective surgical interventions re-presented as emergencies during the course of the lockdown period. This circumstance merits the early and careful consideration put into the delimiting of non-elective surgical interventions. Nevertheless,cancelling elective surgeries reinforced the feeling of insecurity in patients already burdened with steep social changes. Telephone consultations increased during the time of lockdown hinting at a general lack of information about the enforced changes to the healthcare system.

In retrospect, the capacity to treat patients with COVID-19 has never been fully tested by the actual number of admissions. Thus, in addition to declining numbers of emergency cases led to a reduced utilization of hospital resources. Those resources could have been put to good use treating those who refrained from presenting to a hospital due to fear infection and lack of information regarding the COVID-19 situation in our healthcare system.

From an epidemiological point of view, experience with a pandemic of this magnitude is scarce. Based on the assumption that hospital admissions decline despite constant incidence of certain diseases (e.g. brain tumors) an accumulation of cases leading to higher numbers of hospital admissions in the wake of the pandemic are to be expected. Thus, our healthcare system is not only obliged to accommodate enough resources to counter COVID-19 but to be prepared to provide for more than the average share of cases once the pandemic subsides.

## Conclusion

During the present pandemic, it is the incumbent upon all healthcare providers to foster awareness in the neediest of patients, so that essential treatment may be rendered expeditiously, without pointless or harmful delays.

## Author Contributions

**Conceptualization:** Harald Krenzlin, Christoph Bettag, Veit Rohde, Florian Ringel, Naureen Keric.

**Data curation:** Harald Krenzlin, Christoph Bettag, Veit Rohde, Naureen Keric.

**Resources:** Florian Ringel.

**Supervision:** Veit Rohde, Florian Ringel, Naureen Keric.

**Visualization:** Harald Krenzlin.

**Writing – original draft:** Harald Krenzlin, Naureen Keric.

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
