## [Decision Letter · Decision Letter 0]

25 May 2020

PONE-D-20-12668

Involuntary ambulatory triage during the COVID-19 pandemic – a neurosurgical perspective

PLOS ONE

Dear Dr. Krenzlin,

Thank you for submitting your manuscript to PLOS ONE. After careful consideration, we feel that it has merit but does not fully meet PLOS ONE’s publication criteria as it currently stands. Therefore, we invite you to submit a revised version of the manuscript that addresses the points raised during the review process.

We look forward to receiving your revised manuscript.

Kind regards,

Itamar Ashkenazi

Academic Editor

PLOS ONE

Journal Requirements:

2. In the ethics statement in the manuscript and in the online submission form, please provide additional information about the patient records used in your retrospective study, including: a) the date range (month and year) during which the medical records of patients who visited the medical center in Gottingen were reivewed; and b) the source of the medical records analyzed in this work (e.g. the full name of the medical center in Gottingen).

3. To comply with PLOS ONE submission guidelines, in your Methods section, please provide additional information regarding your statistical analyses. For more information on PLOS ONE's expectations for statistical reporting, please see https://journals.plos.org/plosone/s/submission-guidelines.#loc-statistical-reporting.

Reviewers' comments:

Reviewer's Responses to Questions

**Comments to the Author**

1. Is the manuscript technically sound, and do the data support the conclusions?

Reviewer #1: Yes

Reviewer #2: Yes

Reviewer #3: Yes

2. Has the statistical analysis been performed appropriately and rigorously? 

Reviewer #1: Yes

Reviewer #2: Yes

Reviewer #3: Yes

3. Have the authors made all data underlying the findings in their manuscript fully available?

Reviewer #1: Yes

Reviewer #2: Yes

Reviewer #3: Yes

4. Is the manuscript presented in an intelligible fashion and written in standard English?

Reviewer #1: Yes

Reviewer #2: Yes

Reviewer #3: Yes

5. Review Comments to the Author

Reviewer #1: The authors evaluated the impact of the COVID 19 pandemic on admission from Jan 1st to April 19 2020 and compared to previous years. They found significant decline in the emergency admission rate during the timing of lock down.

The authors discussed the decline in all aspect of neurosurgery seen. The impact of this decline was not discussed as the authors discussed they do not have the time.

Additionally there is no mention on whether the reduced admission significantly impacted the overall availability of ICU beds, PPE and man power for treatment of COVID 19 patients.

Given the reduction in pathology that would most likely utilized the most healthcare system such as trauma, is there a balance that can be had?

What is the impact on the patient's whose elective surgery were cancelled,

were any patient who was admitted emergently a patient who was previously thought to be an elective case?

Reviewer #2: This manuscript provides an important snapshot into the impact Covid has made on patient care. One minor revision- recommend reword "attempt to evade exposition to patients " to avoid exposure in the discussion.

Reviewer #3: This is useful and very topical study on decline of neurosurgical admission in time of COVID pandemic. Subject is very important in aplanning how to finish selective 'lockdown' policy in hospitals.

We found it topical and very useful

6. PLOS authors have the option to publish the peer review history of their article (what does this mean?). If published, this will include your full peer review and any attached files.

Reviewer #1: Yes: Kingsley Abode-Iyamah

Reviewer #2: No

Reviewer #3: Yes: Marek Czosnyka

---

## [Author Response · Author response to Decision Letter 0]

1 Jun 2020

Response to the reviewers

-> The manuscript has been revised to adhere to the style requirements of PLOS one.

2. In the ethics statement in the manuscript and in the online submission form, please provide additional information about the patient records used in your retrospective study, including: a) the date range (month and year) during which the medical records of patients who visited the medical center in Gottingen were reivewed; and b) the source of the medical records analyzed in this work (e.g. the full name of the medical center in Gottingen).

-> The methods/patient population section has been amended addressing the participating hospitals properly as The Department of Neurosurgery at the University Medical Center of the Johannes Gutenberg University Mainz and The Department of Neurosurgery at the University Hospital of the Georg August University Göttingen.

-> The date range (month and year) during which the medical records of patients who visited the medical center in Gottingen were reivewed has been update within the same section.

3. To comply with PLOS ONE submission guidelines, in your Methods section, please provide additional information regarding your statistical analyses. For more information on PLOS ONE's expectations for statistical reporting, please see https://journals.plos.org/plosone/s/submission-guidelines.#loc-statistical-reporting.

-> GraphPad Prism is now cited in the statistical analysis section. Further, Tukey’s multiple comparison has been detailed as post hoc test. The used T-test is now specified to comply with PLOS ONE's expectations for statistical reporting.

Reviewer #1: The authors evaluated the impact of the COVID 19 pandemic on admission from Jan 1st to April 19 2020 and compared to previous years. They found significant decline in the emergency admission rate during the timing of lock down.

1. The authors discussed the decline in all aspect of neurosurgery seen. The impact of this decline was not discussed as the authors discussed they do not have the time.

-> Emergency admissions dramatically declined during the nationwide lockdown period. Patients in need of neurosurgical care were admitted delayed or not at all. The overall death rate from March to May, 2020 in Germany is higher than the average of three consecutive years before. It is also higher than the COVID-19 associated mortality and thus hints at the impact of declining hospital admissions. A section discussing the impact of the declining hospital admissions together with the European mortality monitoring activity on average death rates has been added to the discussion section.

2. Additionally there is no mention on whether the reduced admission significantly impacted the overall availability of ICU beds, PPE and man power for treatment of COVID 19 patients.

-> A passage has been included under results stating that the capacity for treatment of COVID 19 patients has never been tested by the number of admitted cases. 

3. Given the reduction in pathology that would most likely utilized the most healthcare system such as trauma, is there a balance that can be had?

-> A new passage has been added to the discussion section explaining the ambiguity of additional ICU capacity, spare work force on one hand and fewer numbers of COVID 19 patients and reduced numbers of emergency admissions other than COVID19 (EAOC19 :) on the other.

4. What is the impact on the patient's whose elective surgery were cancelled,

-> Cancelling elective surgeries reinforced the feeling of insecurity in our patients already burdened with steep social changes. Telephone consultations increased during the time of lockdown hinting at a general lack of information about the changes to our healthcare system. This line of thought has also been added to the discussion section of our manuscript.

5. Were any patient who was admitted emergently a patient who was previously thought to be an elective case?

-> Interestingly, our hospitals have not seen elective surgical interventions presenting as emergencies during the lockdown period. This might partially be due to the careful considerations put into the delimiting of non-elective surgical interventions by the German Society of Neurosurgery (DGNC) and the Professional Association of German Neurosurgeons (BDNC). This circumstance has now been to the discussion section of our manuscript.

Reviewer #2: This manuscript provides an important snapshot into the impact Covid has made on patient care. 

1. One minor revision- recommend reword "attempt to evade exposition to patients " to avoid exposure in the discussion.

-> The wording has been changed accordingly.

Reviewer #3: This is useful and very topical study on decline of neurosurgical admission in time of COVID pandemic. Subject is very important in aplanning how to finish selective 'lockdown' policy in hospitals. We found it topical and very useful

-> We are honored by the appreciation and endorsement of our manuscript.

---

## [Editor Report · Decision Letter 1]

8 Jun 2020

Involuntary ambulatory triage during the COVID-19 pandemic – a neurosurgical perspective

PONE-D-20-12668R1

Dear Dr. Krenzlin,

We’re pleased to inform you that your manuscript has been judged scientifically suitable for publication and will be formally accepted for publication once it meets all outstanding technical requirements.

Kind regards,

Itamar Ashkenazi

Academic Editor

PLOS ONE
---

## [Editor Report · Acceptance letter]

10 Jun 2020

PONE-D-20-12668R1 

Involuntary ambulatory triage during the COVID-19 pandemic – a neurosurgical perspective 

Dear Dr. Krenzlin:

I'm pleased to inform you that your manuscript has been deemed suitable for publication in PLOS ONE. Congratulations! Your manuscript is now with our production department. 

Kind regards, 

on behalf of

Dr. Itamar Ashkenazi 

Academic Editor

PLOS ONE